# ESTIMATING STATISTICAL SIMILARITY BETWEEN PRODUCT DISTRIBUTIONS

## ABSTRACT

We investigate the problem of computing the *statistical* or *total variation (TV) similarity* between distributions $P$ and $Q$, which is defined as $s_{\mathrm{TV}}(P,Q) := 1 - d_{\mathrm{TV}}(P,Q)$, where $d_{\mathrm{TV}}$ is the total variation distance between $P$ and $Q$. Statistical similarity is a basic measure of similarity between distributions with several natural interpretations. We focus on the case when $P$ and $Q$ are products of Bernoulli trials. Recent work has established, somewhat surprisingly, that even for this simple class of distributions exactly computing the TV distance (and hence statistical similarity) is #P-hard. This motivates the question of designing multiplicative approximation algorithms for these computational tasks. It is known that the TV distance computation admits a fully polynomial-time deterministic approximation scheme (FPTAS). It remained an open question whether efficient approximation schemes exist for estimating the statistical similarity between two product distributions. In this work, we affirmatively answer this question by designing an FPTAS for estimating the statistical similarity between two product distributions. To obtain our result, we introduce a new variant of the knapsack problem, which we call multidimensional Masked Knapsack problem, and design an FPTAS to estimate the number of solutions to this problem. This result might be of independent interest.

## 1 INTRODUCTION

The total variation (TV) or statistical distance between distributions $P$ and $Q$ over a common finite sample space $D$, denoted by $d_{\mathrm{TV}}(P,Q)$, is defined as $d_{\mathrm{TV}}(P,Q) := \max_{S \subseteq D}(P(S) - Q(S)) = \frac{1}{2}\sum_{x \in D}|P(x) - Q(x)|$. This distance measure naturally defines a similarity measure which is called *TV similarity* or *statistical similarity*. In particular, the statistical similarity between distributions $P$ and $Q$ is defined as $s_{\mathrm{TV}}(P,Q) := 1 - d_{\mathrm{TV}}(P,Q)$.

The statistical (distance) similarity is a fundamental measure for quantifying the (dis)similarity between probability distributions. It has the following intuitive interpretation: If $s_{\mathrm{TV}}(P,Q) \geq 1 - \varepsilon$, then for any event $E$, its probability with respect to $P$ and $Q$ differs by at most $\varepsilon$. Moreover, the fundamental fact that any process, deterministic or randomized, cannot decrease statistical similarity between two random variables is very useful in module analysis of large systems. Because of these reasons, the notions of statistical distance or similarity between probability distributions has been used in many areas, including probabilistic algorithms Mitzenmacher & Upfal (2005), machine learning Shalev-Shwartz & Ben-David (2014), and information theory Cover & Thomas (2006).

To further motivate the notion, we elaborate on some natural interpretations and equivalent characterizations of statistical similarity between distributions. Interestingly, $s_{\mathrm{TV}}$ can be interpreted by using the notion of a *coupling* between probability distributions. A coupling between probability distributions $P$ and $Q$ is a random variable $(X, Y)$ where $X \sim P$, $Y \sim Q$. An optimal coupling $O = (X, Y)$ is a coupling for which $\mathbf{Pr}_O[X = Y]$ is maximized. It is well known that in optimal case, $s_{\mathrm{TV}}(P,Q) = \mathbf{Pr}_O[X = Y]$. Thus, by computing $s_{\mathrm{TV}}$ between $P$ and $Q$ we can compute the probability that $X$ equals $Y$ under the optimal coupling $O$. Couplings, introduced by Doeblin (1938), have been fundamental in the realms of computer science and mathematics, and have underpinned some of the most seminal results (Lindvall, 2002; Levin et al., 2006; Meyn & Tweedie, 2012).

Moreover, it is known that the minimal total error in hypothesis testing equals the statistical similarity between the underlying distributions (Lehmann & Romano, 2008; Nielsen, 2014). In a similar

vein, there is a connection between statistical similarity and the error of an optimal aggregated predictor (Parisi et al., 2014; Berend & Kontorovich, 2015; Kontorovich, 2024). Consider the following prediction game. A parameter $p_Y \in (0, 1)$ is fixed and a random bit $Y \in \{0, 1\}$ is drawn according to Bernoulli trial with bias $p_Y$, that is, $p_Y = \mathbf{Pr}[Y = 1]$. Conditional on $Y$, the sequence $X_1, X_2, \ldots, X_n$ is drawn i.i.d., where $X_i \in \{0, 1\}$ such that $\mathbf{Pr}[X_i = 1|Y = 1] := \psi_i$ and $\mathbf{Pr}[X_i = 1|Y = 0] := \eta_i$ for some collection of parameters $\psi, \eta \in (0, 1)^n$. The parameters $\psi = (\psi_1, \cdots, \psi_n)$ and $\eta = (\eta_1, \cdots, \eta_n)$ are known as sensitivity and specificity, respectively. An agent who knows the values of $\psi, \eta$ gets to observe the $X_1, \ldots, X_n$ and wishes to infer the most likely $Y$. An optimal predictor $f^{\mathsf{OPT}} : \{0, 1\}^n \to \{0, 1\}$ that minimizes the error probability $\mathbf{Pr}[f^{\mathsf{OPT}}(X) \neq Y]$ was given by Parisi et al. (2014). Kontorovich (2024) showed that the decision rule $f^{\mathsf{OPT}}$ satisfies $\mathbf{Pr}[f^{\mathsf{OPT}}(X) \neq Y] = \frac{1}{2}s_{\mathrm{TV}}(\mathrm{Bern}(\psi), \mathrm{Bern}(\eta))$ whereby $\mathrm{Bern}(\psi)$ is the product of $\mathrm{Bern}(\psi_i)$'s (similarly for $\mathrm{Bern}(\eta)$). Thus the statistical similarity between $\mathrm{Bern}(\psi)$ and $\mathrm{Bern}(\eta)$ precisely captures the error of the optimal predictor.

The above discussion demonstrates that statistical similarity is a fundamental concept with broad applicability across various domains. In this work we focus on the computational aspects of $s_{\mathrm{TV}}$. Recent work has established that exact computation of $s_{\mathrm{TV}}$ is hard. In fact, somewhat surprisingly, Bhattacharyya et al. (2023) showed that it is #P-hard to exactly compute $d_{\mathrm{TV}}$ between two *product distributions*. Thus, as $s_{\mathrm{TV}} = 1 - d_{\mathrm{TV}}$, exactly computing statistical similarity between product distributions is also #P-hard. Hence it is unlikely that there exists an efficient algorithm for this computational task. This motivates the following question:

*Is there an efficient multiplicative approximation algorithm for estimating the statistical similarity between product distributions?*

Recent works of (Bhattacharyya et al., 2023; Feng et al., 2023; 2024) showed that efficient multiplicative approximation algorithms exist for estimating *total variation distance* between product distributions. Since $s_{\mathrm{TV}} = 1 - d_{\mathrm{TV}}$, it might appear that $s_{\mathrm{TV}}$ can be estimated using a $d_{\mathrm{TV}}$ estimation algorithm. However for a multiplicative approximation, this is not the case. That is, it is not possible in general to use an efficient multiplicative approximation algorithm for a function $f$ in order to design an efficient multiplicative approximation algorithm for $1 - f$.

For instance, let $f$ be a function that takes as input a Boolean DNF formula $\phi$ and outputs the probability that a random assignment satisfies $\phi$. It is known that there is a randomized multiplicative approximation algorithm for estimating $f$ Karp et al. (1989). However, a multiplicative approximation algorithm for estimating $1 - f$ implies that all NP-complete problems have efficient randomized algorithms (RP = NP). This is because the complement of a DNF formula is a CNF formula and there is no efficient randomized multiplicative approximation for estimating the acceptance probability of CNF formulas unless RP = NP.

## 1.1 Our Results

To describe our result, we first recall the notion of a fully polynomial-time approximation scheme (FPTAS). An FPTAS $\mathcal{A}$ for $s_{\mathrm{TV}}$ (for product distributions) is a *deterministic polynomial-time* algorithm that takes as input (1) two product distributions $P$ and $Q$ and (2) an accuracy error parameter $0 < \varepsilon$, and outputs a $(1 + \varepsilon)$-multiplicative approximation of $s_{\mathrm{TV}}(P, Q)$. That is, $\mathcal{A}$ outputs a value $v$ so that

$$\frac{s_{\mathrm{TV}}(P, Q)}{(1 + \varepsilon)} \leq v \leq (1 + \varepsilon)s_{\mathrm{TV}}(P, Q).$$

Our main result is an FPTAS that estimates the statistical similarity between two product distributions.

**Theorem 1.** *There is an FPTAS for estimating $s_{\mathrm{TV}}(P, Q)$, where $P, Q$ are arbitrary product distributions over $n$ variables, such that each associated Bernoulli parameter can be encoded using $\ell$ bits. This FPTAS runs in time $O(\mathrm{poly}(\ell, n, 1/\varepsilon))$, whereby $\varepsilon$ is the accuracy error of the FPTAS.*

Our algorithm is obtained by a chain of reductions via two intermediate counting problems that we define next. The first problem is called #MinPMFAtLeast, defined as follows: Given product distributions $P$ and $Q$ over $\{0, 1\}^n$ and $C \geq 0$, compute the number of $x \in \{0, 1\}^n$ such that $\min(P(x), Q(x)) \geq C$. We first show that approximating $s_{\mathrm{TV}}$ between product distributions reduces to approximating #MinPMFAtLeast, as stated below.

**Proposition 2.** *For any $\delta > 0$, computing a $(1 + \delta)$-multiplicative approximation to $s_{\mathrm{TV}}(P, Q)$ for product distributions $P, Q$ can be efficiently reduced to computing a $(1 + \varepsilon)$-multiplicative approximation to polynomially-many $\#\text{MINPMFATLEAST}$ instances over $P$, and $Q$, whereby $\varepsilon = \Omega(\delta)$.*

The second problem that we define is a counting variant of the Knapsack problem which we call multidimensional $\#\text{MASKEDKNAPSACK}$. A multidimensional $\#\text{MASKEDKNAPSACK}$ instance $I$ consists of $m$ standard KNAPSACK instances $K_1, \ldots, K_m$ where each $K_i$ is specified with weights $a_{i,1}, \ldots, a_{i,n}$ and capacity $b_i$. Additionally each $K_i$ is associated with a mask vector $u_i = u_{i,1}, \ldots, u_{i,n} \in \{0, 1\}^n$. An string $x \in \{0, 1\}^n$ is a *solution* to $I$, if $\sum_{j=1}^{n} a_{i,j}(x_j \oplus u_{i,j}) \leq b_i$ for all $1 \leq i \leq m$. (Here $\oplus$ denotes the bitwise XOR operation.) The goal is to compute the number of solutions for a given input instance $I$.

We prove that $\#\text{MINPMFATLEAST}$ (exactly) reduces to multidimensional $\#\text{MASKEDKNAPSACK}$.

**Proposition 3.** $\#\text{MINPMFATLEAST}$ *reduces to multidimensional* $\#\text{MASKEDKNAPSACK}$ *with* $m = 2$.

Finally, we design an FPTAS for multidimensional $\#\text{MASKEDKNAPSACK}$ when $m$ is a constant. This is a general result and might be of independent algorithmic interest.

**Theorem 4.** *There is an FPTAS for multidimensional* $\#\text{MASKEDKNAPSACK}$ *when* $m = O(1)$. *The running time of this FPTAS is* $O\left((n/\varepsilon)^{O(1)}\right) \log W$, *whereby* $\varepsilon$ *is the desired accuracy error and* $W$ *is the maximum total weight among the* MASKEDKNAPSACK *instances.*

## 1.2 RELATED WORK

The computational aspects of TV distance have attracted attention from a complexity theoretic viewpoint, where it has been shown that additive approximations of TV distance between distributions belong to various zero-knowledge classes (Goldreich et al., 1999; Sahai & Vadhan, 2003; Malka, 2008; Dixon et al., 2020; Bouland et al., 2017). In all of these works, the class of distributions considered are distributions samplable by polynomial-size circuits. The work of Sahai & Vadhan (2003) established that additively approximating the TV distance between two distributions that are samplable by Boolean circuits is hard for the complexity class SZK (Statistical Zero Knowledge). Since complexity of additive approximations for $d_{\mathrm{TV}}$ and $s_{\mathrm{TV}}$ are equivalent, the above result holds also for statistical similarity. Goldreich et al. (1999) showed that the problem of deciding whether a distribution samplable by a Boolean circuit is close or far from the uniform distribution is complete for NISZK (Non-Interactive Statistical Zero Knowledge). A recent work of Bhattacharyya et al. (2023) considered much simpler class of distributions. They showed that $(a)$ exactly computing the TV distance between product distributions is $\#$P-complete, and $(b)$ multiplicatively approximating the TV distance between Bayes nets is NP-hard.

Regarding algorithmic aspects, Bhattacharyya et al. (2020) designed efficient algorithms to additively approximate the TV distance between distributions that are efficiently samplable and efficiently computable. Feng et al. (2023) designed a fully polynomial-time randomized approximation scheme (FPRAS) for estimating the TV distance between two arbitrary product distributions. Interestingly, their work used couplings to devise the algorithm. More recently, Feng et al. (2024) gave an FPTAS for the same task.

## 1.3 PAPER ORGANIZATION

We present some background material in Section 2. We prove Proposition 2 in Section 3, Proposition 3 in Section 4, Theorem 4 in Section 5, and Theorem 1 in Section 6. Finally, we conclude in Section 7 with some problems. In Appendix A, we present the pseudocode for all of our procedures.

## 2 PRELIMINARIES

We use $[n]$ to denote the set $\{1, \ldots, n\}$. We will use $\log$ to denote $\log_2$. The following notion of a deterministic approximation algorithm is important for this work.

**Definition 5.** A function $f : \{0,1\}^* \to \mathbb{R}$ admits a *fully polynomial-time approximation scheme (FPTAS)* if there is a *deterministic* algorithm $\mathcal{A}$ such that for every input $x$ (of length $n$) and $\varepsilon > 0$, the algorithm $\mathcal{A}$ outputs a $(1 + \varepsilon)$-multiplicative approximation to $f(x)$, i.e., a value that lies in the interval $[f(x)/(1 + \varepsilon), (1 + \varepsilon)f(x)]$. The running time of $\mathcal{A}$ is polynomial in $n$ and $1/\varepsilon$.

## 2.1 PRODUCT DISTRIBUTIONS

A Bernoulli distribution with parameter $p$ is denoted by $\mathrm{Bern}(p)$. A *product distribution* is a product of independent Bernoulli distributions. A product distribution $P$ over $\{0,1\}^n$ can be described by $n$ Bernoulli parameters $p_1, \ldots, p_n$ where each $p_i \in [0, 1]$ is the probability that the $i$-th coordinate equals 1 (such a $P$ is usually denoted by $\mathrm{Bern}(p_1, \ldots, p_n)$ or $\bigotimes_{i=1}^n \mathrm{Bern}(p_i)$). We define $\ell$ to be such that each Bernoulli parameter $p_i$ encountered in this work can be represented by using (at most) $\ell$ bits. For any $x \in \{0,1\}^n$, the probability of $x$ with respect to the product distribution $P$ is given by

$$P(x) = \prod_{i \in S_x} p_i \prod_{i \in [n] \setminus S_x} (1 - p_i) \in [0, 1] \,,$$

whereby $S_x \subseteq [n]$ is such that $i \in S_x$ if and only if $x_i = 1$.

## 2.2 TOTAL VARIATION DISTANCE AND STATISTICAL SIMILARITY

The following notion of distance between distributions is central in this work.

**Definition 6.** For distributions $P, Q$ over a sample space $D$, the *total variation (TV) distance between $P$ and $Q$* is

$$d_{\mathrm{TV}}(P, Q) := \max_{S \subseteq D}(P(S) - Q(S)) = \frac{1}{2} \sum_{x \in D} |P(x) - Q(x)| = \sum_{x \in D} \max(0, P(x) - Q(x)) \,.$$

The *TV similarity or statistical similarity between $P$ and $Q$* is $s_{\mathrm{TV}}(P, Q) := 1 - d_{\mathrm{TV}}(P, Q)$.

We present a characterization of $s_{\mathrm{TV}}$ that we have used in this work. We present its proof for completeness.

**Proposition 7** (Scheffé's identity, see also (Tsybakov, 2009))**.** *Let $P, Q$ be distributions over a sample space $D$. Then*

$$s_{\mathrm{TV}}(P, Q) = \sum_{x \in D} \min(P(x), Q(x)) \,.$$

*Proof.* We have that

$$s_{\mathrm{TV}}(P, Q) = 1 - \sum_{x \in D} \max(0, P(x) - Q(x))$$

$$= \sum_{x \in D} P(x) + \sum_{x \in D} \min(0, Q(x) - P(x))$$

$$= \sum_{x \in D} \min(P(x), P(x) + Q(x) - P(x)) = \sum_{x \in D} \min(P(x), Q(x)) \,. \qquad \square$$

## 2.3 COUNTING PROBLEMS

A function $f$ from $\{0,1\}^*$ to nonnegative integers is in the class #P if there is a polynomial-time nondeterministic Turing machine $M$ so that for any $x$ the value of $f(x)$ is equal to the number of accepting paths of $M(x)$.

### 2.3.1 #MASKEDKNAPSACK

Let us first remind the reader the standard #KNAPSACK problem: Given weights $a_1, \ldots, a_n$ and capacity $b$, compute the number of sets $S \subseteq [n]$ such that $\sum_{i \in S} a_i \leq b$. For a KNAPSACK instance with weights $a_1, \ldots, a_n$ and a capacity $b$, its *total weight* is $\sum_{i=1}^n a_i + b$. It is a folklore result that #KNAPSACK is #P-hard.

In this paper, we study the following "masked" notion of KNAPSACK.

**Definition 8** (#MASKEDKNAPSACK). #MASKEDKNAPSACK is the following counting problem. We are given a KNAPSACK instance $K$, defined by a set of weights $a_1, \ldots, a_n$, a capacity $b$, and a mask $u = u_1, \ldots, u_n \in \{0,1\}^n$. We say that $x$ is a *solution* to $K$ (in symbols, $x \in S$) if $\sum_{j=1}^n a_j \left( x_j \oplus u_j \right) \leq b$. The computational goal is to count the number of solutions, that is, the size of $S$. Moreover, the sum $\sum_{i=1}^n a_i + b$ is called the *total weight* of the instance.

It is a straightforward observation that #MASKEDKNAPSACK is #P-hard, since one may reduce #KNAPSACK to #MASKEDKNAPSACK by setting the mask $u$ to be an all-zeroes string. We focus on a particular kind of multidimensional #KNAPSACK that is defined over $m$ MASKEDKNAPSACK instances.

**Definition 9** (Multidimensional #MASKEDKNAPSACK). Consider MASKEDKNAPSACK instances $K_1, \ldots, K_m$, whereby $K_i$ is defined by a set of weights $a_{i,1}, \ldots, a_{i,n}$, a capacity $b_i$, and a mask $u_i = u_{i,1}, \ldots, u_{i,n} \in \{0,1\}^n$. We have that $x$ is a solution to $K_i$ (in symbols, $x \in S_i$) if $\sum_{j=1}^n a_{i,j} \left( x_j \oplus u_{i,j} \right) \leq b_i$. The output is the size of $S = \bigcap_{i=1}^m S_i$.

## 3 REDUCTION FROM STATISTICAL SIMILARITY TO #MINPMFATLEAST

We prove Proposition 2.

*Proof of Proposition 2.* Let $P$ and $Q$ be two product distributions. We will reduce

$$s_{\mathrm{TV}}(P, Q) = \sum_{x \in \{0,1\}^n} \min(P(x), Q(x))$$

to a collection of polynomially many #MINPMFATLEAST instances over $P$ and $Q$.

Let $m_{\min}$ and $m_{\max}$ denote the minimum and maximum nonzero values of $\min(P(x), Q(x))$ over all $x$. By our assumption on the bit representation of the parameters $p_i, q_i$, we get that $m_{\min} \geq m_0 := \left( 2^{-\ell} \right)^n = 2^{-\ell n}$. Moreover, $m_{\max} \leq 1$. Let $V \geq 1$ be a number so that $\min(P(x), Q(x))/m_0 \leq V$ for all $x$. Therefore, $V \leq m_{\max}/m_0 \leq 1/m_0 = 2^{\ell n}$. In fact, let us set $V := 2^{\ell n}$. Let $Y_x := \min(P(x), Q(x))/m_0$ and note that $Y_x$ lies in $[1, V)$.

We will divide the interval $[1, V)$ into sub-intervals that are multiples of $(1 + \varepsilon)$ for some $\varepsilon$ that is within a linear factor of $\delta$ which we will fix later. More precisely, let

$$[1, V) = \bigcup_{i=0}^{u-1} \left[ (1 + \varepsilon)^i, (1 + \varepsilon)^{i+1} \right)$$

be a set of sub-intervals for $0 \leq i \leq u - 1 = \left\lceil \log_{1+\varepsilon} V \right\rceil - 1 \leq \mathrm{poly}(\ell, n, 1/\varepsilon)$. For any $0 \leq i \leq u - 1$, let $n_i$ denote the number of $x \in \{0,1\}^n$ such that $Y_x$ is in $\left[ 1, (1 + \varepsilon)^i \right)$. That is,

$$n_i := \left| \left\{ x \mid Y_x \in \left[ 1, (1 + \varepsilon)^i \right) \right\} \right|.$$

Let the average contribution of $Y_x$ in the range $\left[ (1 + \varepsilon)^{i-1}, (1 + \varepsilon)^i \right)$ be $B_i$. That is, $B_i := \sum Y_x/(n_i - n_{i-1})$, where the sum is over all $Y_x$ in the interval $\left[ (1 + \varepsilon)^{i-1}, (1 + \varepsilon)^i \right)$. Then we have the following equation:

$$\frac{s_{\mathrm{TV}}(P, Q)}{m_0} = n_1 B_1 + (n_2 - n_1) B_2 + (n_3 - n_2) B_3 + \cdots + (n_u - n_{u-1}) B_u. \tag{1}$$

Since $(1 + \varepsilon)^{i-1} \leq B_i < (1 + \varepsilon)^i$, the following estimate $d$ is a $(1 + \varepsilon)$-approximation of the RHS of Equation (1):

$$d := n_1(1 + \varepsilon) + (n_2 - n_1)(1 + \varepsilon)^2 + (n_3 - n_2)(1 + \varepsilon)^3 + \cdots + (n_u - n_{u-1})(1 + \varepsilon)^u. \tag{2}$$

By reorganizing the terms of Equation (2), we get

$$d = \left( (1 + \varepsilon)^u - (1 + \varepsilon)^{u-1} \right) (n_u - n_{u-1})$$
$$+ \left( (1 + \varepsilon)^{u-1} - (1 + \varepsilon)^{u-2} \right) (n_u - n_{u-2}) + \cdots + (1 + \varepsilon) n_u. \tag{3}$$

Therefore it suffices to estimate $n_u - n_j$ for every $1 \leq j \leq u - 1$. (We know that $n_u = 2^n$.) By definition, $t_j := n_u - n_j$ counts the number of $x \in \{0,1\}^n$ such that $Y_x \geq (1+\varepsilon)^j$. Note that

$$Y_x \geq (1+\varepsilon)^j \Leftrightarrow \min(P(x), Q(x)) \geq (1+\varepsilon)^j\, m_0.$$

That is, $t_j$ counts the number of $x \in \{0,1\}^n$ such that $\min(P(x), Q(x)) \geq (1+\varepsilon)^j\, m_0$. If we estimate each $t_j$ up to a $(1+\varepsilon)$-multiplicative approximation, this in turn would give us a $(1+\varepsilon)$-multiplicative approximation for $d$ by Equation (3), and for that matter a $(1+\varepsilon)^2$-multiplicative approximation for $s_{\mathrm{TV}}(P,Q)$ by Equation (1). Hence, if we set $\varepsilon := \Omega(\delta/2)$ so that $(1+\varepsilon)^2 \leq (1+\delta)$, we get the desired approximation ratio of $(1+\delta)$ for $s_{\mathrm{TV}}(P,Q)$. $\qquad\square$

# 4    REDUCTION FROM #MINPMFATLEAST TO MULTIDIMENSIONAL #MASKEDKNAPSACK

We prove Proposition 3.

*Proof of Proposition 3.* Let $P$ and $Q$ be two product distributions with Bernoulli parameters $p_1, \ldots, p_n$ and $q_1, \ldots, q_n$, respectively. The goal is to show that #MINPMFATLEAST, that is, computing $|\{x \in \{0,1\}^n \mid \min(P(x), Q(x)) \geq C\}|$, can be written as an instance of #MASKEDKNAPSACK.

We first give some notation and definitions that are necessary for the proof. Let

$$a_i := \max\left(\frac{p_i}{1-p_i}, \frac{1-p_i}{p_i}\right) \qquad \text{and} \qquad b_i := \min(p_i, 1-p_i),$$

and

$$c_i := \max\left(\frac{q_i}{1-q_i}, \frac{1-q_i}{q_i}\right) \qquad \text{and} \qquad d_i := \min(q_i, 1-q_i).$$

For any $x \in \{0,1\}^n$ define sets $T_P$ and $T_Q$ as follows:

$$T_P(x) := \left\{i \in [n] \mid p_i \geq \frac{1}{2},\ x_i = 1 \quad \text{or} \quad p_i \leq \frac{1}{2},\ x_i = 0\right\},$$

$$T_Q(x) := \left\{i \in [n] \mid q_i \geq \frac{1}{2},\ x_i = 1 \quad \text{or} \quad q_i \leq \frac{1}{2},\ x_i = 0\right\}.$$

For all $x \in \{0,1\}^n$, let $S_x$ be such that $i \in S_x$ if and only if $x_i = 1$ (that is, $x$ is the characteristic vector of $S_x$).

We require the following claim.

**Claim 10.** *It is the case that*

$$P(x) = \prod_{i \in S_x} p_i \prod_{i \notin S_x} (1 - p_i) = \left(\prod_{i=1}^n b_i\right)\left(\prod_{i \in T_P} a_i\right),$$

$$Q(x) = \prod_{i \in S_x} q_i \prod_{i \notin S_x} (1 - q_i) = \left(\prod_{i=1}^n d_i\right)\left(\prod_{i \in T_Q} c_i\right).$$

The proof of Claim 10 is straightforward, and it is based on appropriately rearranging the factors of the PMFs of $P$ and $Q$. Thus the inequalities $P(x) \geq C$ and $Q(x) \geq C$ are equivalent to

$$\left(\prod_{i=1}^n b_i\right)\left(\prod_{i \in T_P(x)} a_i\right) \geq C \quad \text{and} \quad \left(\prod_{i=1}^n d_i\right)\left(\prod_{i \in T_Q(x)} c_i\right) \geq C,$$

or

$$\prod_{i \notin T_P(x)} a_i \leq \frac{\left(\prod_{i=1}^n a_i\right)\left(\prod_{i=1}^n b_i\right)}{C} \qquad \text{and} \qquad \prod_{i \notin T_Q(x)} c_i \leq \frac{\left(\prod_{i=1}^n c_i\right)\left(\prod_{i=1}^n d_i\right)}{C},$$

since $a_i, c_i \geq 1$ for all $i$. In order to make the product a sum, we can take $\log$ on both sides, yielding

$$\sum_{i \in [n] \setminus T_P(x)} \log a_i \leq \log \frac{\left(\prod_{i=1}^n a_i\right)\left(\prod_{i=1}^n b_i\right)}{C}, \quad \sum_{i \in [n] \setminus T_Q(x)} \log c_i \leq \log \frac{\left(\prod_{i=1}^n c_i\right)\left(\prod_{i=1}^n d_i\right)}{C}.$$

At this point, the expressions look *similar* to KNAPSACK constraints. While we do not know how to cast them as standard KNAPSACK constraints, we can frame them as MASKEDKNAPSACK constraints as follows.

Let $y_1(x)$ be the characteristic vector of $[n] \setminus T_P(x)$ and $y_2(x)$ be the characteristic vector of $[n] \setminus T_Q(x)$. Then the above inequalities become

$$\sum_{i=1}^n (\log a_i) \, y_1(x)_i \leq \log \frac{\left(\prod_{i=1}^n a_i\right)\left(\prod_{i=1}^n b_i\right)}{C},$$

$$\sum_{i=1}^n (\log c_i) \, y_2(x)_i \leq \log \frac{\left(\prod_{i=1}^n c_i\right)\left(\prod_{i=1}^n d_i\right)}{C}.$$

Define masks $u_P$ and $u_Q$ corresponding to $P$ and $Q$ as follows: $u_P = u_{P,i}, \ldots, u_{P,n}$ is such that $u_{P,i} = 1$ if and only if $p_i \geq 1/2$, and $u_Q = u_{Q,i}, \ldots, u_{Q,n}$ is such that $u_{Q,i} = 1$ if and only if $q_i \geq 1/2$. Then from the definition of $T_P, T_Q$ and $u_P, u_Q$ the above inequalities can be written as

$$\sum_{i=1}^n (\log a_i) \, (x_i \oplus u_{P,i}) \leq \log \frac{\left(\prod_{i=1}^n a_i\right)\left(\prod_{i=1}^n b_i\right)}{C},$$

$$\sum_{i=1}^n (\log c_i) \, (x_i \oplus u_{Q,i}) \leq \log \frac{\left(\prod_{i=1}^n c_i\right)\left(\prod_{i=1}^n d_i\right)}{C}.$$

Thus, for an instance $P, Q, C$ of #MINPMFATLEAST we can construct two instances $I_P$ and $I_Q$ of #MASKEDKNAPSACK where $I_P$ is specified by the weights $\log a_1, \ldots, \log a_n$, capacity $\log \frac{\left(\prod_{i=1}^n a_i\right)\left(\prod_{i=1}^n b_i\right)}{C}$, and the mask $u_P$, and $I_Q$ is specified by weights $\log c_1, \ldots, \log c_n$, capacity $\log \frac{\left(\prod_{i=1}^n c_i\right)\left(\prod_{i=1}^n d_i\right)}{C}$, and the mask $u_Q$, so that for all $x \in \{0,1\}^n$ it is the case that $\min(P(x), Q(x)) \geq C$ if and only if $x$ is a solution to $I_P$ and a solution to $I_Q$. Finally, note that this reduction runs in time linear in $n$. This completes the proof. $\qquad \square$

## 5 COUNTING MULTIDIMENSIONAL MASKEDKNAPSACK SOLUTIONS

### 5.1 BACKGROUND ON BRANCHING PROGRAMS

We first fix some notation and terminology. A $(W, n)$-*branching program* is a branching program of width $W$ over $n$ Boolean input variables. A *read-once branching program (ROBP)* is a branching program whereby each input variable is accessed only once. A *monotone $(W, n)$-ROBP* is a $(W, n)$-ROBP such that in each of its layers $L$, the nodes of $L$ are totally ordered under some relation $\prec$, and whenever $u \prec v$ for some nodes $u$ and $v$ it is the case that the set of partial accepting paths that start at $u$ are a subset of the set of partial accepting paths that start at $v$.

Given a branching program $M$ and a string $z$, the notation $M(z)$ denotes the output ("accept"/"reject") of $M$ on input $z$. An *implicit description* of a monotone ROBP is a description according to which one can efficiently check the relative order of two nodes under $\prec$ (within any layer), and given a node $u$ one can efficiently compute its neighbors.

The following notion of small-space sources was introduced by Kamp et al. (2011).

**Definition 11** (Kamp et al. (2011)). A *width-$w$ small-space source* is described by a $(w, n)$-branching program $D$ with an additional probability distribution $p_v$ on the outgoing edges associated with vertices $v \in D$. Samples from the source are generated by taking a random walk on $D$ according to the $p_v$'s and outputting the labels of the edges traversed.

We require the following useful claims by Gopalan et al. (2010). Claim 12 is an application of dynamic programming.

**Claim 12** (Gopalan et al. (2010)). *Given a ROBP $M$ of width at most $W$ and a small-space source $D$ of width at most $S$, it is the case that $\mathbf{Pr}_{x \sim D}[M(x) = 1]$ can be computed exactly in time $O(nSW)$.*

**Claim 13** (Gopalan et al. (2010)). *Given a $(W, n)$-ROBP $M$, the uniform distribution over $M$'s accepting inputs, $\{x \mid M(x) = 1\}$ is a width $W$ small-space source.*

We further require the following important result from Gopalan et al. (2010).

**Theorem 14** (Gopalan et al. (2010)). *Given a monotone $(W, n)$-ROBP $M$, $\delta > 0$, and a small-space source $D$ over $\{0, 1\}^n$ of width at most $S$, there exists an $(O(n^2 S/\delta), n)$-monotone ROBP $M_0$ such that for all $z$, it is the case that $M(z) \leq M_0(z)$ and*

$$\Pr_{z \sim D}[M(z) = 1] \leq \Pr_{z \sim D}[M_0(z) = 1] \leq (1 + \delta) \Pr_{z \sim D}[M(z) = 1].$$

*Moreover, given an implicit description of $M$ and a description of $D$, $M_0$ can be constructed in deterministic time $O(n^3 S(S + \log W) \log(n/\delta)/\delta)$.*

The main take-away of Theorem 14 is that the number of accepting paths of $M_0$ (under the distribution $D$) approximates the number of accepting paths of $M$ (under the distribution $D$), and moreover $M_0$ has small width.

## 5.2 Proof of Theorem 4

We prove Theorem 4. To this end, we first show Lemma 15. This lemma is based on the Dyer's rounding scheme in the context of standard #KNAPSACK.

**Lemma 15** (Rounding). *Given a collection of MASKEDKNAPSACK instances KNAPSACKs $K_1, \ldots, K_m$, each over $n$ variables and with a total weight of at most $W$, and solution sets $S_1, \ldots, S_m$, respectively, we can deterministically in time $O(n^3 \log W)$ construct new MASKEDKNAPSACK instances $K_1', \ldots, K_m'$ with solution sets $S_1', \ldots, S_m'$, respectively, each with a total weight of at most $O(n^3)$, such that $S_i \subseteq S_i'$ for all $1 \leq i \leq m$ and*

$$\left| \bigcap_{i=1}^m S_i' \right| \leq n^m \left| \bigcap_{i=1}^m S_i \right|.$$

*Proof.* Let

$$S_i := \left\{ x \in \{0, 1\}^n \mid \sum_{j=1}^n a_{i,j} (x_j \oplus u_{i,j}) \leq b_i \right\},$$

whereby $0 \leq a_{i,1} \leq \cdots \leq a_{i,n} \leq b_i$. Let $k_i$ be such that $a_{i,j} \leq b_i/n$ for $j \leq k_i$ and either $k_i = n$ or $a_{i,k_i+1} > b_i/n$. Let $C_i := \left\{ z, u_{i,k_i+1}, \ldots, u_{i,n} \mid z \in \{0, 1\}^{k_i} \right\}$. If $x \in C_i$, then

$$\sum_{j=1}^n a_j (x_j \oplus u_{i,j}) \leq \sum_{j=1}^n a_j \leq k_i b_i/n \leq b_i$$

and so $x \in S_i$. That is, $C_i \subseteq S_i$. Let now $\alpha_{i,j} := \lfloor n^2 a_{i,j}/b_i \rfloor$ and $\delta_{i,j} := n^2 a_{i,j}/b_i - \alpha_{i,j}$, such that $0 \leq \delta_{i,j} < 1$. Let also

$$S_i' := \left\{ x \in \{0, 1\}^n \mid \sum_{j=1}^n \alpha_{i,j} (x_j \oplus u_{i,j}) \leq n^2 \right\}$$

and $S := \bigcap_{i=1}^m S_i$, $S' := \bigcap_{i=1}^m S_i'$. We will prove that $|S| \leq |S'| \leq n^m |S|$. Let us first prove that $|S| \leq |S'|$. Let $x \in S$. Then for all $i$ we have

$$\sum_{j=1}^n \alpha_{i,j} (x_j \oplus u_{i,j}) \leq (n^2/b_i) \sum_{j=1}^n a_{i,j} (x_j \oplus u_{i,j}) \leq (n^2/b_i) b_i = n^2,$$

so $x \in S_i'$ and therefore $x \in S'$. Thus $S \subseteq S'$ and so $|S| \leq |S'|$.

Let us now show that $|S'| \leq n^m |S|$. To this end, let $L_i := \{j \mid a_{i,j} \leq b_i/n\}$. For $x \in S' \setminus S$, let $I(x) := \{i \mid x \in S'_i \setminus S_i\}$. For every $i \in I(x)$, there exists $p_i(x) \notin L_i$ such that $x_{p_i} \oplus u_{i,p_i} = 1$ and $\alpha_{i,p_i(x)} \geq n$. Otherwise, $x \in C_i \subseteq S_i \subseteq S'_i$. (If there exist more than one such integer, take $p_i(x)$ to be the smallest.) Construct $f(x) = y$ by $y_{p_i(x)} = 0$ for $i \in I(x)$ and $y_j = x_j$ otherwise. Then for any $x \in S' \setminus S$, with $y = f(x)$, we have

$$\sum_{j=1}^{n} a_{i,j} \left( y_j \oplus u_{i,j} \right) = \frac{b_i}{n^2} \sum_{j=1}^{n} \left( \alpha_{i,j} + \delta_{i,j} \right) \left( y_j \oplus u_{i,j} \right)$$

$$= \frac{b_i}{n^2} \left( \sum_{j=1}^{n} \alpha_{i,j} \left( y_j \oplus u_{i,j} \right) + \sum_{j=1}^{n} \delta_{i,j} \left( y_j \oplus u_{i,j} \right) \right)$$

$$= \frac{b_i}{n^2} \left( \sum_{j=1}^{n} \alpha_{i,j} \left( x_j \oplus u_{i,j} \right) - \alpha_{i,p_i(x)} \left( x_{p_i} \oplus u_{i,p_i(x)} \right) + \sum_{j=1}^{n} \delta_{i,j} \left( y_j \oplus u_{i,j} \right) \right)$$

$$= \frac{b_i}{n^2} \left( \sum_{j=1}^{n} \alpha_{i,j} \left( x_j \oplus u_{i,j} \right) - \alpha_{i,p_i(x)} + \sum_{j=1}^{n} \delta_{i,j} \left( y_j \oplus u_{i,j} \right) \right)$$

$$\leq \frac{b_i}{n^2} \left( n^2 - n + n \right) = b_i.$$

That is, $f(x) \in S_i$ and so $f(x) \in S$. Hence $f(S') = S$. The inverse mapping changes some set of coordinates $P$ with $0 \leq |P| \leq m$, so

$$\left| f^{-1}(y) \right| \leq 1 + n + \binom{n}{2} + \cdots + \binom{n}{m} \leq n^m.$$

That is, $|S| \leq n^m |S'|$. Therefore $|S'| \leq \left| f^{-1}(S) \right| \leq n^m |S|$. $\qquad\square$

We may now prove Theorem 4 by using Lemma 15.

*Proof of Theorem 4.* We will appeal to the techniques of Gopalan et al. (2010). First, we will apply Lemma 15 to obtain MASKEDKNAPSACK instances $K'_1, \ldots, K'_m$, each with a total weight of at most $O(n^3)$, and solution sets $S'_1, \ldots, S'_m$, respectively.

Let $D$ be the uniform distribution over the set $S' := \bigcap_{i=1}^{m} S'_i$ and observe that by Claim 13 $D$ can be generated by an explicit $O(n^{3m})$ space source. For $1 \leq i \leq m$, let $M^i$ be a $(W, n)$-ROBP exactly computing the indicator function for $S_i$. Let $\delta = O(\varepsilon/(m(n+1)^m))$ to be chosen later. For every $1 \leq i \leq m$, by Theorem 14 we can explicitly in time $n^{O(m)}(\log W)/\delta$ construct a $(n^{O(m)}/\delta, n)$-ROBP $M^i_r$ such that

$$\mathbf{Pr}\left[ M^i_r(x) \neq M(x) \right] \leq \delta.$$

Define $M$ such that $M(x) := \bigwedge_{i=1}^{m} M^i_r(x)$ for any $x$. Then $M$ is a $\left( n^{O(m^2)}/\delta^m, n \right)$-ROBP. By a union bound,

$$\mathbf{Pr}_{x \sim D}\left[ M(x) \neq \bigwedge_{i=1}^{m} M^i(x) \right] \leq m\delta.$$

On the other hand, by Theorem 14,

$$\mathbf{Pr}_{x \sim D}\left[ \bigwedge_{i=1}^{m} M^i(x) = 1 \right] \geq 1/(n+1)^m.$$

Therefore, by setting $\delta := \varepsilon/(2m(n+1)^m)$, we get

$$\mathbf{Pr}_{x \sim D}[M(x) = 1] \leq \mathbf{Pr}_{x \sim D}\left[ \bigwedge_{i=1}^{m} M^i(x) = 1 \right] \leq (1 + \varepsilon)\,\mathbf{Pr}_{x \sim D}[M(x) = 1].$$

Thus, $p := \mathbf{Pr}_{x \sim \{0,1\}^n}[x \in S'] \mathbf{Pr}_{x \sim D}[M(x) = 1]$ is a $(1 + \varepsilon)$-multiplicative approximation to the fraction of solutions to all constraints

$$\mathbf{Pr}_{x \sim \{0,1\}^n}\left[\bigwedge_{i=1}^{m} M^i(x) = 1\right] = \mathbf{Pr}_{x \sim \{0,1\}^n}[x \in S'] \mathbf{Pr}_{x \sim D}\left[\bigwedge_{i=1}^{m} M^i(x) = 1\right].$$

The result now follows since we can compute $p$ in time $(n/\delta)^{O(m^2)}$ using Claim 12, as $D$ is a small-space source of width $O(n^{3m})$ and $M$ has width $(n/\delta)^{O(m^2)}$. □

## 6 ESTIMATING STATISTICAL SIMILARITY

We now prove Theorem 1 by combining the previous results.

*Proof of Theorem 1.* By Proposition 2, the $(1 + \delta)$-multiplicative approximation of $s_{\mathrm{TV}}(P, Q)$ reduces to the $(1 + \varepsilon)$-multiplicative approximation of polynomially-many #MINPMFATLEAST instances over $P, Q$, namely $t_1, \ldots, t_k$, for $\varepsilon = \Omega(\delta/2)$ and $k = \mathrm{poly}(\ell, n)$. By Proposition 3, the instances $t_1, \ldots, t_k$ can be reduced to multidimensional #MASKEDKNAPSACK for $m = 2$. Using Theorem 4, we can estimate each $t_j$ up to a $(1 + \varepsilon)$-multiplicative approximation in deterministic polynomial time.

The running time of this algorithm is polynomial in $\ell, n, 1/\delta$ because we ran a polynomial-time approximation algorithm for multidimensional #MASKEDKNAPSACK polynomially many times. In particular, the running time is $\mathrm{poly}(\ell, n, 1/\varepsilon) \cdot O\left((n/\delta)^{O(1)}\right) \log W = O\left((\ell \cdot n/\delta)^{O(1)}\right)$ (since $W = \mathrm{poly}(n)$, by Lemma 15). □

## 7 CONCLUSION

We have given a simple FPTAS for estimating the statistical similarity between product distributions. We reiterate that the known FPTAS for TV distance Feng et al. (2024) does not in general yield an FPTAS for statistical similarity. In fact, we use different techniques than that of Feng et al. (2024) to design the FPTAS for statistical similarity. Our algorithm is based on a reduction to a new knapsack counting problem which we call (multidimensional) #MASKEDKNAPSACK which might be of independent interest. Extending our results to more general classes of distributions and establishing lower bounds is a promising and significant research direction. Finally, we believe that a complexity-theoretic study of functions $f$ in #P with range in $[0, 1]$, for which there are approximation schemes for both $f$ and $1 - f$, is significant.

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

## A  PSEUDOCODE

We present the pseudocode of our algorithms, in reverse order.

We present the pseudocode for Theorem 4 in Algorithm 1.

---

**Algorithm 1** The pseudocode for Algorithm 1.

---

**Require:** $m$ instances of MASKEDKNAPSACK, specified by weights $\{a_i\}_{i=1}^m$, whereby $a_i = a_{i,1}, \ldots, a_{i,n}$, mask vectors $\{u_i\}_{i=1}^m$, whereby $u_i = u_{i,1}, \ldots, u_{i,n}$, capacities $b_1, \ldots, b_m$, and an accuracy error parameter $\varepsilon$.

**Ensure:** The output $p$ is an $(1 + \varepsilon)$-estimate to multidimensional #MASKEDKNAPSACK.

1: {By parsing the input, we can compute $m$ and $n$.}
2: **for** $i \leftarrow 1, \ldots, m$ **do**
3:     **for** $j \leftarrow 1, \ldots, n$ **do**
4:         $\alpha_{i,j} \leftarrow \lfloor n^2 a_{i,j}/b_i \rfloor$
5:     **end for**
6: **end for**
7: $S' \leftarrow \{0,1\}^n$
8: **for** $i \leftarrow 1, \ldots, m$ **do**
9:     {It is the case that $S_i := \left\{ x \in \{0,1\}^n \mid \sum_{j=1}^n a_{i,j} \left( x_j \oplus u_{i,j} \right) \leq b_i \right\}$.}
10:     Compute $M^i$
11:     {$M^i$ is a $(W, n)$-ROBP exactly computing the indicator function for $S_i$.}
12:     Compute $M_r^i$
13:     {$M_r^i$ is a $\left( n^{O(m)}/\delta, n \right)$-ROBP that is a *rounding* of $M^i$, as given by Theorem 14.}
14:     $S_i' \leftarrow \left\{ x \in \{0,1\}^n \mid \sum_{j=1}^n \alpha_{i,j} \left( x_j \oplus u_{i,j} \right) \leq n^2 \right\}$
15:     {The set $S_i'$ can be computed by dynamic programming in time polynomial in $n$.}
16:     $S' \leftarrow S' \cap S_i'$
17: **end for**
18: $M \leftarrow \bigwedge_{i=1}^m M_r^i$
19: $p_D \leftarrow \mathbf{Pr}_{x \sim D}[M(x) = 1]$
20: {The probability $p_D$ can be computed by Claim 12.}
21: $p_{S'} \leftarrow |S'|/2^n$
22: {Note that $p_{S'} = \mathbf{Pr}_{x \sim \{0,1\}^n}[x \in S']$.}
23: $p \leftarrow p_D \cdot p_{S'}$
24: **return** $p$

---

We present the pseudocode for Proposition 2 in Algorithm 2.

We present the pseudocode for Proposition 2 in Algorithm 3.

**Algorithm 2** The pseudocode for Proposition 3.

**Require:** Product distributions $P, Q$ through their Bernoulli parameters $p_1, \ldots, p_n, q_1, \ldots, q_n$, and a parameter $C$.

**Ensure:** The output $I$ is an instance of the multidimensional $\#\text{MASKEDKNAPSACK}$ problem for $m = 2$.

1: {By parsing the input, we can compute $n$.}
2: $\Pi_a \leftarrow 1$
3: $\Pi_b \leftarrow 1$
4: $\Pi_c \leftarrow 1$
5: $\Pi_d \leftarrow 1$
6: **for** $i \leftarrow 1, \ldots, n$ **do**
7:     $a_i \leftarrow \max\left(\frac{p_i}{1-p_i}, \frac{1-p_i}{p_i}\right)$
8:     $b_i \leftarrow \min(p_i, 1 - p_i)$
9:     $c_i \leftarrow \max\left(\frac{q_i}{1-q_i}, \frac{1-q_i}{q_i}\right)$
10:     $d_i \leftarrow \min(q_i, 1 - q_i)$
11:     $\Pi_a \leftarrow \Pi_a \cdot a_i$
12:     $\Pi_b \leftarrow \Pi_b \cdot b_i$
13:     $\Pi_c \leftarrow \Pi_c \cdot c_i$
14:     $\Pi_d \leftarrow \Pi_d \cdot d_i$
15:     **if** $p_i \geq 1/2$ **then**
16:       $u_P, i \leftarrow 1$
17:     **else**
18:       $u_P, i \leftarrow 0$
19:     **end if**
20:     **if** $q_i \geq 1/2$ **then**
21:       $u_Q, i \leftarrow 1$
22:     **else**
23:       $u_Q, i \leftarrow 0$
24:     **end if**
25: **end for**
26: $C_P \leftarrow \log(\Pi_a \Pi_b / C)$
27: $C_Q \leftarrow \log(\Pi_c \Pi_d / C)$
28: $I_P \leftarrow \left((\log a_i)_{i=1}^n, u_P, C_P\right)$
29: $I_Q \leftarrow \left((\log c_i)_{i=1}^n, u_Q, C_Q\right)$
30: $I \leftarrow (I_P, I_Q)$
31: **return** $I$

**Algorithm 3** The pseudocode for Proposition 2.

---

**Require:** Product distributions $P, Q$ through their Bernoulli parameters $p_1, \ldots, p_n, q_1, \ldots, q_n$, and an accuracy error parameter $\delta$.

**Ensure:** The output $d$ is an $(1 + \delta)$-estimate of $s_{\mathrm{TV}}(P, Q)$.

1: {By parsing the input, we can compute $n$.}
2: {We define $\langle \cdot \rangle_2$ to be a function that maps any number $x$ to its (standard) binary representation in $\{0, 1\}^*$.}
3: $\ell \leftarrow 0$
4: **for** $i \leftarrow 1, \ldots, n$ **do**
5:     $\ell \leftarrow \max(\ell, |\langle p_i \rangle_2|, |\langle q_i \rangle_2|)$
6: **end for**
7: $m_0 \leftarrow 2^{-\ell n}$
8: $V \leftarrow 2^{\ell n}$
9: $u \leftarrow \log_{1+\varepsilon} V$
10: {Note that $u \leq \mathrm{poly}(\ell, n, 1/\varepsilon)$.}
11: $n_u \leftarrow 2^n$
12: $d \leftarrow (1 + \varepsilon) \, n_u$
13: **for** $k \leftarrow 2, \ldots, u$ **do**
14:     $t_k \leftarrow \text{Algorithm 1}\Big(\text{Algorithm 2}\Big(P, Q, (1 + \varepsilon)^k \, m_0\Big), \delta/2\Big)$
15:     $d \leftarrow d + \Big((1 + \varepsilon)^k - (1 + \varepsilon)^{k-1}\Big) t_k$
16: **end for**
17: **return** $d$

---

