# OpenReview forum: "Estimating Statistical Similarity Between Product Distributions"
_ICLR.cc/2025/Conference — Submitted to ICLR 2025_

### Official Review · Reviewer_Se8d · 2024-10-28

**Soundness:** 2
**Presentation:** 2
**Contribution:** 2
**Rating:** 3
**Confidence:** 3

**Summary:**

This paper is about estimating statistical similarity between product distributions P and Q, which is defined by one minus total variation distance between P and Q, i.e. d_TV(P,Q). Estimating the exactly total variation distance between product distributions is #P hard, and hence estimating the statistical similarity is also #P hard. On the approximation side, there is an FPTAS for estimation the TV distance, however that would not give an approximation for the statistical similarity, as a multiplicative approximation for a function f does not give a multiplicative approximation for 1-f, and there are examples that such approx does not exist for 1-f.

This paper provides an FPTAS for estimating statistical similarity between product distributions. They use two reductions, the first reduce the problem to #MinPMFAtleast problem, which given C, asks for the number of inputs x such that min(P(x),Q(x)) >= C. Then they reduce this problem to a variant of the Knapsack problem that they introduce, called multidimensional #MaskedKnapsack. Finally they give an FPTAS for this variant of Knapsack.

**Strengths:**

They provide enough motivations, the proofs seem simple.

**Weaknesses:**

There are a lot of typos in the manuscript, including a lot of inequalities that should be reversed, which didn’t let me verify the proofs. I will pose my questions in the questions section. Besides this, I don’t see this result as being very relevant to ICLR, both in strength, and in the fact that it might be more relevant to theory conferences like SODA or ICALP.

Some typos:
line 57: there “is” a connection “between” statistical similarity “and" the error of an optimal …
Line 69: Finally, the “statistical” similarity can be…
Line 78:: Hence it is unlikely that an efficient algorithm “exists” …
Line 287: remove “have”
Line 365: start “at” u

**Questions:**

1) In proof of proposition 2, there are many typos which essentially make the proof wrong but I think that they are all fixable, so please provide the correct proof.
Line 242: this expression for m_min does not make any sense to me. I think you probably mean  to use an inequality, not an equality.
Like 245, you define V to be $min(P(x),Q(x))/m_{min}\le V$. I think you mean to use $\ge$, since then you get an upper bound for V by using the fact that min(P(x),Q(x)). But you also need to show that $V\ge 1$, which means that the definition of V is not for all x, it is for x where min(P(x),Q(x)) is non-zero. This subtlety shows itself agin on line 272 where you say $n_u=2^n$.
Line 259, the definition of B_i should be the sum of all Y_x where Y_x is in the $[(1+\epsilon)^{i-1},(1+\epsilon)^i)$ interval. Please specify this

2) Please specify the running time of your algorithm.

3) Please compare your techniques to the work before especially the work that provides a FPTAS for TV distance. How original are your techniques?

---

> ### Author Response · Authors · 2024-11-22
>
> Dear Reviewer,
>
> thank you very much for taking the time to review our paper.
>
> Let us now respond to your comments:
> 1. We took care of all of the typos you pointed out. Thank you!
> 2. We have fixed the proof of Proposition 2.
> 3. We have included the running time of our FPTAS. Please see Theorem 1.
> 4. We have read and understood the previous work that gives an FPTAS for TV distance. However, it was not clear to us how to adapt it to the setting of statistical similarity. In particular, the estimator idea that they propose for estimating $d_{\rm TV}$ seems to be not suitable for estimating $s_{\rm TV} = 1 - d_{\rm TV}$. To the best of our knowledge, our techniques are original.

---

### Official Review · Reviewer_THL6 · 2024-11-03

**Soundness:** 3
**Presentation:** 3
**Contribution:** 3
**Rating:** 6
**Confidence:** 4

**Summary:**

The authors study the computation of the total variation similarity, s_TV (P,Q) = 1-d_TV(P,Q), between product distributions. The problem is  #P-hard following the analogous result about the computation of the total variation distance d_TV(P,Q). For the latter an FPTAS is known to exists, which does not directly carry over to the computation of s_TV. The paper shows an FPTAS for s_TV. This is done by reducing the problem in two steps to the computation of a variant of multiple Knapsack problems.

**Strengths:**

Since the total variation similarity is extensively used and the only known approximations are for the distance, the paper closes the gap between the two problems for the case of general product distributions, which also closes the gap of the complexity landascape for the total similarity.
The reduction used to the masked multi knapsack is neat and novel and might be of independent interest

**Weaknesses:**

Approximation is for product distribution only, but on the other hand this gives an complete picture of the complexity for this case.

**Questions:**

In the statement of Proposition 2, it is not clear what is meant by an approximation of a polynomially many #MinPMFAtLeast instances. Do you mean "... to the computing a (1+e)-approximation of polynomially many instances of...  "?

It seems to me that the formula for m_{\min}--- on page 5, the 2nd equation in disply in the proof od Proposition 2---provides a lower bound on the minimum non-zero value of min(P(x), Q(x)) over all x, rather than the  exact minimum non-zero value of min(P(x), Q(x)) over all x, which has been given as the definition of  m_{\min}.
I do not think this affects the result, but, if I am write, the definition should be correct.

Specific Comments
In the statement of Lemma 15, the dimension parameter n should be explicitly defined.

---

> ### Author Response · Authors · 2024-11-22
>
> Dear Reviewer,
>
> thank you very much for taking the time to review our paper.
>
> Let us now respond to your comments:
> 1. You are right, this is what we intended to write. We have made the appropriate edits.
> 2. We have removed the formula for $m_{\min}$, as it is not really needed in the paper. What is important is that $m_{\min}$ is at least $m_0 := 1 / (2^{\ell n})$, whereby $\ell$ is the number of bits that suffice to represent any Bernoulli parameter $p_i$ or $q_i$.
> 3. We have explicitly defined $n$ in Lemma 15. Thank you!

---

### Official Review · Reviewer_TU1g · 2024-11-03

**Soundness:** 3
**Presentation:** 2
**Contribution:** 2
**Rating:** 3
**Confidence:** 3

**Summary:**

This paper studies the hardness of computing the statistical similarity and shows that there exists an FPTAS for approximating the statistical similarity between product distributions. For two distributions $P$ and $Q$, the statistical similarity between $P$ and $Q$ is defined as $s_{TV}(P, Q) = 1-d_{TV}(P, Q)$, where $d_{TV}(.,.)$ is the total variation distance. In this work,

1- The authors first reduced approximating $s_{TV}$ to the #MinPMFAtLeast problem in which, given two product distributions $P$ and $Q$ over $\{0,1\}^n$ and $C\ge 0$, computes the number of $x\in\{0,1\}^n$ such that $\min\{P(x), Q(x)\}\ge C$.
2- Then, they reduced #MinPMFAtLeast to the 2-dimensional #MaskedKnapsack problem in which, given two instances $K_1$ and $K_2$ specified by weights $a_{i,1}, \ldots, a_{i,n}$, capacity $b_i$, and mask vectors $u_i=(u_{i,1}, \ldots, u_{i,n})$ for $i\in{1,2\}$, counts the number of solutions $x=(x_1, \ldots, x_n)$ such that $\sum_{j=1}^n a_{i,j}(x_j\oplus u_{i,j}) \le b_i for $i=1,2$.
3- Next, they show that there exists an FPTAS for $O(1)$-dimensional #MaskedKnapsack problem.
4- Finally, they approximately solved #MaskedKnapsack using techniques from [Gopalan et al (2010)].

**Strengths:**

- The paper shows that there exists an FPTAS for approximating the statistical similarity between product distributions.

**Weaknesses:**

- The paper does not motivate the problem well. There are some vague mentions of machine learning, probabilistic algorithms, information theory, Markov chains, and hypothesis testing with references to general source books on the topics, and it is not clear to me at all what are the actual applications of statistical similarity between product distributions. See my questions below.
- There is an assumption that the probabilities of the Bernoulli random variables can be represented using poly(n) bits. This assumption is not stated in the main Theorem.
- Unless there are specific use cases of relative approximation of statistical similarity that I am unaware of, I think this work would be more suited for the theory venues.

**Questions:**

1. Are there any papers or work specifically using (a relative approximation of) statistical similarity between product distributions?
2. The following are the typos I could find, but there are probably more:

Line 56: there a connection -> there is a connection

Line 70: statistically similarity -> statistical similarity

Line 70: related well-known -> related to well-known

Line 287: product distributions with have -> product distributions which have

Line 364: that start $u$ -> that start at $u$

Line 365: the the -> the

Line 417: $k$ -> $k_i$ (?)

---

> ### Author Response · Authors · 2024-11-22
>
> Dear Reviewer,
>
> thank you very much for taking the time to review our paper.
>
> Let us now respond to your comments:
> 1. The following paper
>     >Aryeh Kontorovich:
> Aggregation of expert advice, revisited. CoRR abs/2407.16642 (2024)
>
>     uses statistical similarity between product distributions.
> 2. We have striven to improve the motivation of statistical similarity.
> 3. We have updated our algorithm to **not** need the assumption on the representation size of the Bernoulli parameters.
> 4. We took care of all of the typos you pointed out. Thank you!

---

> > ### Comment · Reviewer_TU1g · 2024-11-26
> >
> > Thank you for your response.
> >
> > The paper you mentioned establishes theoretical bounds on the statistical similarity between product distributions but does not appear to apply them directly (unless I’ve overlooked something). I would be interested to know if they are utilizing these bounds in a machine learning-related application.
> >
> > That said, I still believe the paper is better suited for theory conferences, as noted by another reviewer as well.

---

### Official Review · Reviewer_ySz5 · 2024-11-03

**Soundness:** 3
**Presentation:** 2
**Contribution:** 2
**Rating:** 5
**Confidence:** 3

**Summary:**

This paper considers the problem of estimating the statistical similarity between distributions that are product of n Bernoulli trials. Statistical similarity is simply 1-(total variation distance). This paper proposes a fully polynomial time approximation algorithm for the problem. It is noteworthy that there are additive and relative approximations for the total variation distance that run in polynomial time. I believe that an additive approximation of TV distance translates to an additive approximation of statistical similarity. However, this is not the case for relative approximations of TV distance and statistical similarity and there is no straightforward relationship between them. Thus, although closely related, relative approximations of TV distance cannot be directly used for solving relative approximation of statistical similarty.

The main approach of the paper is to reduce the problem of estimating statistical similarity to a variant of the knapsack problem (masked version) and then designing a relative approximation for this variant.

**Strengths:**

TV distance as well as statistical similarity are relevant to ML community and progress on approximation algorithms for these measures are important.

**Weaknesses:**

* The authors do not explicitly compare their techniques with the existing techniques for additive and relative approximation of the TV distance. They compare them at the level of results but not at the level of techniques. Ideally, I would like to see an explanation of why prior work for TV distance cannot be used for solving for statistical similarity.
* The algorithm requires a polynomial bit representation for probabilities. This is a somewhat surprising requirement for approximation algorithms (usually these kinds of requirements are mostly seen in weakly polynomial exact algorithms). Why is it hard to eliminate this requirement?
* Presentation can be significantly improved.
* Implementation and testing would have made the paper stronger

I will raise my score if most of my concerns are addressed.

**Questions:**

Apart from addressing the weaknesses, I would also like to ask if there is any literature on the case where P and Q are product of one-dimensional continuous distributions and what we have is only samples from P and Q. This problem is also important in ML and I’m not aware of the state-of-the art for relative and additive approximation for this problem. Could you shed some light on the results for this variation of your problem?

---

> ### Author Response · Authors · 2024-11-22
>
> Dear Reviewer,
>
> thank you very much for taking the time to review our paper.
>
> Let us now respond to your comments:
> 1. We have read and understood the previous work that gives an FPTAS for TV distance. However, it was not clear to us how to adapt it to the setting of statistical similarity. In particular, the estimator idea that they propose for estimating $d_{\rm TV}$, seems to be not suitable for estimating $s_{\rm TV} = 1 - d_{\rm TV}$.
> 2. We have updated our algorithm to **not** need the assumption on the representation size of the Bernoulli parameters.
> 3. We have striven to improve the presentation. Thank you!
> 4. Implementation and testing of the proposed FPTAS is one of our future work goals :)
> 5. To the best of our knowledge, we do not know of such a reference.

---

### Official Review · Reviewer_gyNW · 2024-11-04

**Soundness:** 3
**Presentation:** 3
**Contribution:** 3
**Rating:** 6
**Confidence:** 4

**Summary:**

The work addresses the computational problem of estimating the statistical similarity ($s_{TV}$) between two product distributions. Computing the exact $s_{TV}$ is #P-hard. The authors aim to develop an FPTAS for estimating $s_{TV}$ between arbitrary product distributions. To achieve this goal, they introduce a new variant of the knapsack problem called the multidimensional #MASKEDKNAPSACK and estimate the number of solutions to this problem. Their result is obtained by a chain of reductions: from the $s_{TV}$ estimation problem to the #MINPMFATLEAST problem, and then to the multidimensional #MASKEDKNAPSACK problem, which is supported by rigorous mathematical proofs.

**Strengths:**

This paper addresses an important problem by attempting to develop an FPTAS for estimating statistical similarity between product distributions, which is a #P-hard problem. The introduction of the multidimensional #MASKEDKNAPSACK problem looks like a novel extension of the classic knapsack problem. The authors provide solid proof for the proposed reductions and approximations. The chain of reductions is logically sound. I recognize the theoretical contribution of this paper.

**Weaknesses:**

In general, I tend to accept this paper. I think the major concern is that this paper claims to develop an FPTAS for estimating $s_{TV}$ between product distributions but does not provide an explicit algorithm or pseudocode. Without such an example, it's hard to assess the practicality of the proposed schema.

**Questions:**

- Can the authors provide explicit algorithmic steps or pseudocode for the proposed FPTAS?

---

> ### Author Response · Authors · 2024-11-22
>
> Dear Reviewer,
>
> thank you very much for taking the time to review our paper.
>
> We have included pseudocode for all of our algorithms in our appendix (see Appendix A).

---

### Meta-Review · Area_Chair_8r3J · 2024-12-20

**Metareview:**

While the reviewers all felt that this work had merit, the ultimate consensus is that the novelty and relevance to ICLR are limited in comparison to other submissions this year. The paper could be strengthened by better motivating the need for a multiplicative approximation to statistical similarity, especially given the existence of additive approximation methods, and multiplicative approximations to distance (a seemingly better motivated problem). An experimental evaluation (ideally demonstrating an application) would strengthen the paper as well. That said, we enjoyed reading this paper, and wish the authors the best of luck in resubmitting to another venue.

**Additional Comments On Reviewer Discussion:**

There was no significant reviewer discussion -- the reviewers largely understood the paper and the authors mostly acknowledged small changes to fix clarity issues.

---

### Decision · Program_Chairs · 2025-01-22

Reject